# Electrochemical Corrosion Behavior and Mechanical Properties of Nanocrystalline Ti–6Al–4V Alloy Induced by Sliding Friction Treatment

**DOI:** 10.3390/ma12050760

**Published:** 2019-03-05

**Authors:** Jinwen Lu, Wei Zhang, Wangtu Huo, Yongqing Zhao, Wenfang Cui, Yusheng Zhang

**Affiliations:** 1Advanced Materials Research Central, Northwest Institute for Nonferrous Metal Research, Xi’an 710016, China; zhangwei-key@126.com (W.Z.); huowangtu@126.com (W.H.); zhaoyq@163.com (Y.Z.); 2Key Laboratory for Anisotropy and Texture of Materials, Northeastern University, Shenyang 110819, China; cuiwf@126.com

**Keywords:** Ti–6Al–4V alloy, nanostructure, passive film, corrosion resistance, tensile properties

## Abstract

A nanograined (NG) layer with an average grain size of less than 100 nm has been successfully prepared on a Ti–6Al–4V sheet surface by sliding friction treatment (SFT). The electrochemical corrosion/passive behavior and mechanical properties of an NG Ti–6Al–4V sheet were examined in this study. A bi-layer passive film that consisted of an outer TiO_2_-rich layer and an inner Al_2_O_3_-rich layer was formed on either an NG or coarse-grained (CG) surface. The improved corrosion was mainly caused by the enhanced stability and thickness of the passive layer. Tensile experiments were carried out to evaluate the mechanical properties at ambient temperature. The NG Ti–6Al–4V sample exhibited the high yield strength (956 MPa) with a moderate elongation of 8%. These superior comprehensive properties demonstrated its potential as a biomedical material.

## 1. Introduction

Ti–6Al–4V alloy was originally developed for aerospace application, and was also the first titanium alloy used as a biomaterial for orthopedic and dental bioimplants, owing to its well-known combination properties, including high specific strength, excellent workability, and biocompatibility [1,2]. Moreover, Ti–6Al–4V alloy exhibits an excellent corrosion resistance that benefited from the formation of the compact rutile TiO_2_ layer [3,4]. However, it has gradually lost its dominance as the most viable orthopedic alloy, because the metal ions (such as V^3+^, V^5+^, and Al^3+^) resulted from passive layer dissolution and wearing have a great toxic effect during long-term performance [5,6]. Therefore, significantly improved corrosion resistance is needed to satisfy the demand for biomedical application. Recent research studies [7,8] for Ti–6Al–4V alloy have verified that enhancing the electrochemical stability was an alternative approach to reduce the release of toxic alloying elements by passive film dissolution.

It is well-known that nanograined (NG) alloy, with its small grain sizes and high-density grain boundaries, possesses a higher electrochemical stability and corrosion resistance than the corresponding coarse-grained (CG) material [9]. Nowadays, there are many research works attempting to interpret the corrosion mechanism of NG Ti, but different results were obtained. Some studies [10,11] reported that nanocrystalline would provide many active sites to form a protectively oxide film, thus enhancing the corrosion resistance and reducing the ion release. However, the literature [12,13] indicated that the protective passive layer formed on the NG surface contained more oxygen and metal ions (such as Ti, Al, and Nb, etc.), which resulted in the rapid growth of the passive layer. Besides, the structure of the passive layer on the NG surface is also affected by the matrix microstructure [10,12]. Thus, it is considered that a compact passive layer plays a crucial role in blocking the release of ions and improving corrosion resistance. However, the mechanism for the compactness and stability improvement for the passive layer induced by the NG microstructure is still not well understood. This study tries to establish a corresponding corrosion model to analyze the passive process and corrosion resistance mechanism. 

In recent years, a new surface modification technique, sliding friction treatment (SFT), was developed with potential industrial application. By means of this technology, a clean and thick nanocrystalline surface layer on a large dimensional bulk metal sheet can be achieved with a good surface quality [14]. For AZ31 Mg alloy subjected to SFT as an example, an NG surface layer has been fabricated with a thickness of up to 70 μm and a surface roughness of 0.6 μm [15]. The present study intends to investigate the effect of nanocrystallization by SFT on the electrochemical corrosion behavior and mechanical properties of Ti–6Al–4V alloy, involving the growth behavior of passive film dependent on the grain size. Furthermore, the CG Ti–6Al–4V alloy is also examined as control.

## 2. Experimental

### 2.1. Material Preparation 

The material in this study was a Ti–6Al–4V sheet with the thickness of 2.4 mm. The chemical composition measured by chemical analysis was tested as following (in wt.%): 5.41 Al, 4.03 V, 0.08 C, 0.002 H, and 0.005 O, with the balance Ti. Before SFT, the polished Ti–6Al–4V sheet was annealed at 800 °C for 1 h to eliminate the stress concentration effect and gain a homogeneous CG microstructure. The detailed information about the SFT setup and procedures in this study were reported in Ref. [16]. The SFT experiments were conducted with a sliding velocity of 0.2 m/s, normal loads of 300 N, an amplitude of 50 mm, and a duration of 100 cycles.

### 2.2. Microstructure Characterization

The phase identification was carried out by X-ray diffraction (XRD, D8 ADVANCE, Bruker, Karlsruhe, Germany), which operated at 30 kV and 30 mA. The microstructure for the NG sample was performed by transmission electron microscope (TEM, JEM-2100, JEOL, Tokyo, Japan) used at a voltage of 200 kV. The cross-sectional TEM thin foils were obtained by cutting from the corresponding surface NG layer, and then mechanically grinding the foils on the un-SFTed sides to obtain a thin plate of about 20 μm in thickness. The fabricated plates were then electropolished using the twin-jet technique operated at a voltage of 50 V, which was performed in a solution of 10% perchloric acid and methanol at a temperature of −40 °C. The grain size was obtained by the calculation used in the linear intercept method in SEM and TEM images. The microstructure for the CG sample was examined using optical microscopy (OM, Leica MPS 30, Leica, Wizla, Germany) and scanning electron microscope (SEM, JSM-6460, Tokyo, Japan), and the grain sizes were obtained by means of the quantitative image analyzer software with OM images.

### 2.3. Electrochemical Measurements

Electrochemical tests were carried out by IM6 Zahner-electrik Gmbhelectrochemical work station (Zenniom, Kronach, Germany), and each test was performed in simulated body fluid (SBF) solution at 37 ± 1 °C. The traditional three-electrode system was employed using a saturated calomel electrode (SCE) as a reference electrode and a platinum wire as the counter electrode, while the tested samples with a certain exposed area (10 mm × 10 mm) were used as the working electrode. Before the electrochemical test, the sample was connected to a copper wire, and then embedded in an epoxy cold-mounting resin. The surfaces exposed to the electrolyte were polished with different emery paper (150–2000 grade) to gain a mirror-like plane, and then cleaned ultrasonically in acetone, ethanol, and distilled water for 10 min, and dried under a cold air stream. The composition of SBF solution is as follows [17]: 8.035 g/L of NaCl; 0.355 g/L of NaHCO_3_; 0.225 g/L of KCl; 0.321 g/L of K_2_HPO_4_·3H_2_O; 0.311g/L of MgCl_2_·6H_2_O; 0.291 g/L of CaCl_2_; 0.072 g/L of Na_2_SO4, and 6.118 g/L of tris(hydroxymethyl)aminomethane (TRIS). The pH was adjusted to 7.4 by the addition of HCl (1 M) or NaOH (1 M) solution. Before each electrochemical test, SBF solution was statically placed for 10 min under an ambient environment, and was removed after the test for each sample.

Before all the electrochemical tests, the sample surface was initially reduced potentiostatically at –1.0 V for 300 s to remove the air-formed oxide layer, and then kept in the SBF solution until a stable potential was obtained (the stabilized passive layer was also formed simultaneously in this process). In this study, the anodic potential of NG Ti–6Al–4V alloy is about 0.7 V, and the CG alloy is only about 0.1 V; the anodic potential of both reached their pitting potential. The potentiostatic polarization data was analyzed by using the stochastic theory in Ref [18]. The open circuit potential (OCP) measurement was conducted for 30 min from the sample immersing into SBF solution. The potentiodynamic polarization (PDP) test was achieved after 30 min of immersion in the range from −1 V to 2 V with a scan rate of 0.5 mV/s. The corrosion potential (*E*_corr_) was analyzed by Tafel slope extrapolation, and the passive current density (*I*_pp_) was determined from the passive zone where the current density remained approximately constant. An electrochemical impedance spectroscopy (EIS) test with a 10-mV amplitude signal was carried out in the frequencies range from 10^5^ Hz to 10^−2^ Hz at the respective OCP. The Mott–Schottky plot with the applied frequency of 1 kHz was measured in the potential range from −0.5 to 1 V (versus SCE). For the potentiostatic polarization measurement, the variation of current with time was measured at 0.5 V_SCE_. For all the electrochemical tests, the SBF solution of 500 ml was changed for each sample, and was repeated three times as the proposed American Society for Testing Materials (ASTM) standard to ensure accuracy.

### 2.4. Surface Characterization 

The surface morphology was observed by using SEM (JSM-6460, Japan) after electrochemical measurement. X-ray photoelectron spectrometer (XPS, ESCAKAB-250Xi, Thermo Fisher, Waltham, USA) was performed to obtained the chemical composition and thickness of passive layer on both of NG and CG samples. All XPS spectrums were conducted by the parameters of standard peaks, and the data was analyzed using XPSPEAK4.1 software (Thermo Fisher, Waltham, USA).

### 2.5. Immersion Tests

Immersion tests were implemented in SBF solution at 37 ± 1 °C controlled by a thermostatic water bath. The cuboid samples (10 mm × 10 mm × 4 mm) were cut and mechanically polished; then, the samples were cleaned ultrasonically with acetone. After 10 days of immersion, SEM was used to observe the corroded surface, and the chemical composition for immersion surface was also measured by energy-dispersive spectrometer (EDS, Surface scanning, JEOL, Tokyo, Japan) with planar scan mode (the scan area is around 200 μm × 200 μm).

### 2.6. Mechanical Tests

Dog bone-shaped sample (a gauge length and gauge width were 12 mm and 4 mm, respectively) along the SFT direction was machined using an electro discharge machining technique. The thickness of the NG tensile sample was 2.25 mm, which was slightly less than that of the original CG samples (2.40 mm) due to some wear and tear by the SFT. Moreover, it is noted that that double surfaces for tensile samples were prepared by SFT to obtain a unique microstructure with a central CG core sandwiched between both sides of NG layers. Mechanical tests were performed on the Instron598X universal material testing machine, the strain rate with 10^−3^ s^−1^ was selected until fracture. The fracture surface after tensile test was observed by SEM (JSM-6460, Japan).

## 3. Results

### 3.1. Phase Constitutions and Microstructure 

The XRD patterns (Figure 1a) show that only the α phase was detected for the NG and CG samples. It is found that the Bragg peak of the NG sample was broadened significantly compared to its CG form, which is mainly attributed to the formation of a nanocrystalline structure and lattice distortion induced by SFT. During plastic deformation, the dislocation sources (such as Frank–Read sources) become activated, and some dislocations and vacancies occurred in the surface deformation layer. With the continuous increase of the strain and strain rate by SFT, the dislocation strain fields can interact with each other, which makes some atoms in the crystal deviate from the original equilibrium position, resulting in lattice distortion and grain refinement. In the XRD patterns (Figure 1a), the main diffraction peaks of the SFT-treated (NG) sample were considerably broadened. Therefore, the lattice distortion and different microstructures featured dislocation tangles, dislocation walls, and lamella-shaped subgrains that were mainly closer to the surface of the NG layer.

As shown in the OM micrographs of Figure 1b, the CG microstructure is presented by equiaxed α grain (lighter) and a spot of intergranular β phase (darker). The volume percentage of the equiaxed α phase of around 5 to 10 μm in diameter reached approximately 87.50% (Figure 1c). As shown in the TEM images of the NG microstructure in Figure 1d,e, the NG sample consisted of α grains of an average size smaller than 100 nm in the surface layer, with the NG layer thickness of ~20 μm. The elected area electron diffraction (SAED) pattern (inset Figure 1d) shows discontinuous diffused diffraction rings. It indicates that the size of the CG microstructure has been refined into the nanoscale, and some grain boundaries are in a non-equilibrium state due to the accumulation of many dislocations around the boundaries. 

### 3.2. Electrochemical Corrosion Behavior

#### 3.2.1. OCP Measurements

OCP curves are shown in Figure 2a; these display the relationship between the electrode potential and immersion time. The OCP values (*E*_ocp_) for both samples rise gradually to reach the steady state. The initial increase in the *E*_ocp_ is related to the formation of the surface passive layer. The *E*_ocp_ of the NG sample exhibits a rapidly increase during the first 100 s, and then keeps a relatively stable value, indicating the formation of a protective passive layer. For the CG sample, more immersion time is needed to form the passive film. Moreover, it is noticed that a much higher *E*_ocp_ is observed for the NG sample relative to its CG form, implying that a more stable passive layer is formed due to surface nanocrystallization. 

#### 3.2.2. PDP Curves

Typical PDP tests were carried out to investigate the passive film layer formed on the NG and CG surfaces, and the curves are shown in Figure 2b. It is obvious that the NG and CG samples exhibit apparent self-passivation behavior. In order to further understand the protection effect of the passive layer, the Tafel extrapolation method is conducted to calculate corrosion parameters (including *E*_corr_ and *I*_pp_) from PDP plots, and the results are presented in Table 1. It is found that the NG sample had a nobler *E*_corr_ (−0.641 V) than the CG one, which is consistent with their *E*_ocp_ values. Moreover, when the current density (*I*_pp_) reached 25.387 μA/cm^2^, the NG sample entered the passivation region, and the *I*_pp_ for the CG sample needed to arrive at 26.280 μA/cm^2^ to enter the passivation region. These polarization results indicate that the NG sample exhibited a higher tendency to form a stable passive film layer. 

#### 3.2.3. EIS Tests

EIS tests were carried out to further study the electrochemical characteristics, and the EIS plots and simulated curves for the NG and CG samples are shown in Figure 2c,d. The Nyquist curves show a depressed semicircle in the entire frequency range, and the total impedance of the NG sample is also higher than that of the CG one. Furthermore, the NG sample exhibits an increase of the capacitive loop in the Nyquist plot. The further depressed capacitive semicircles are associated with the charge transfer process in the electrode/electrolyte interface. The increase of the semicircle radius indicates an enhanced stability of passive film for corrosion resistance [19,20]. Therefore, the larger capacitive semicircles’ diameter and the increased phase angle all imply that a more protective passive layer was obtained on the NG surface.

The equivalent circuit with two time constants is shown and inserted in Figure 2d to fit the impedance; the fitting quality is evaluated by using the chi-squared value of about 10^−4^ (in Table 2), which consists of two parallel combination circuits of *R*_p_*Q*_p_ and *R*_b_*Q*_b_, in series with *R*_s_. Here, *R*_s_ represents electrolyte resistance; *R*_p_ and *Q*_p_ represent the film resistance and constant phase element (*CPE*) of the outer layer; and *R*_b_ and *Q*_b_ represent the film resistance and constant phase element (*CPE*) of the inner barrier layer [21]. *n*_1_ and *n*_2_ are the exponential coefficient. *CPE* is a non-ideal capacitance element that has a relation with the roughness of the electrode surface, which can be obtained with the method in Ref. [22]. In this equivalent circuit, the passive film presents a bi-layered structure, which agrees well with the results by Bolat [23] and Schulte [24].

The fitting values of the mentioned parameters for both the NG and CG samples have been presented in Table 2, and the variation of the total resistance *R*_p_ + *R*_b_ with the testing time at *E*_ocp_ is shown in Figure 2c. It is found that the resistances of the barrier layer (*R*_b_) for both the NG and CG samples are higher by about 30 to 150 times in comparison to that of the outer layer (*R*_p_), which means that the protection is mainly dependent on the inner barrier layer. The total resistance *R*_p_ + *R*_b_ obviously increases with the refinement of the α grain, and decreases as the erosion time increases, which contributes to the resistive change occurring at the metal/film and film/solution interfaces. In addition, the NG sample exhibits lower capacitance (*Q*_b_) and higher passive film resistance (*R*_b_) than the CG one (Table 2), which reveals the formation of the thickened passive layer and the enhancement of the corrosion resistance.

### 3.3. Characterization of Passive Film

Figure 3a,b show SEM surface morphologies of NG and CG Ti–6Al–4V samples after polarization measurements. Both samples exhibit the similar electrochemical corrosion morphologies. The surfaces of NG and CG samples are all possibly covered by a passive layer without obvious pitting. In order to further understand the improved corrosion resistance for the NG sample, potentiostatic polarization tests, Mott–Schottky curves, and XPS analysis have been conducted to provide the structure (the thickness and composition), compactness, and stability of the passive layer in this section. 

#### 3.3.1. Potentiostatic Polarization Test

The potentiostatic polarization (PP) curve is used to reflect passive film compactness. The initial decrease of current density (*I*) is associated with the stability of a protective layer on the sample surface. With the prolonged polarization, *I* will decay continuously, and the relationship between the time (*t*) and potentiostatic current density can be expressed by the formula [25], *I* = 10^−(*A* + *k*lg*t*)^, where *A* is the constant, and *k* is the slope of the double-log plot. According to Ref [26], k→−1 indicates the formation of a compact passive layer, while k→−0.5 means the growth of a porous film. Figure 3c shows the double-log plots for both samples in SBF solution. During the initial stage of approximately 20 s, the values of *k* for the NG and CG samples are −0.65 and −0.13, which reveals a porous passive film formed on both sample surfaces. However, there is an obvious difference for the *k* values between both samples at the last stage—including changes from k = −0.65 to k = −1.11 for the NG sample and from k = −0.13 to k = −0.38 for the CG sample—indicating that a more compact passive layer ultimately forms on the NG surface.

#### 3.3.2. Mott–Schottky Analysis

The passive layer formed on the titanium alloy surface exhibits semiconductor properties, with metal oxides as n-type semiconductors [27]. The Mott–Schottky (MS) plots based on the measured electrode capacitance with the applied potential are used to analyze the structural evolutions of the passive layer at the electrochemical interface. The linear relation between 1/*C*^2^ and *E* is expressed as a Mott–Schottky equation [28]. The donor density (*N_D_*) can be determined by the slope of *C^−2^* and *E* in MS plots. 

Figure 3d shows the MS curves for passive layer on the NG and CG surfaces in SBF solution. In the potential range of around −0.5 to 1 V_SCE_, a linear relationship can be observed between 1/*C*^2^ and *E*; the slopes in those ranges suggested that the passive layers formed on both samples are all n-type semiconductor slopes. At low film formation potential, it could be found that there exists break points, and this phenomenon is related to surface roughness and non-uniform donor distribution within the passive layer [29]. The values (*N*_D_) of the NG sample under higher film form potential ranges (0.5 to 2.0 V_SCE_) were 7.107 × 10^19^ cm^−3^, whereas 8.698 × 10^19^ cm^−3^ in the same potential ranges were also obtained for the CG one. A lower *N*_D_ for the NG sample indicates the formation of a highly protective passive film.

#### 3.3.3. XPS Analysis of Passive film

Figure 4 shows the high-resolution XPS spectra of all the elements in passive layers for the NG and CG samples at different sputtering depths. The passive films of both samples have the same composition, including Ti, Al, V, and O. According to the peak strength of Ti 2p_3/2_, Al 2p_3/2_, and V 2p_3/2_ in high-resolution XPS signals, irrespective of grain size, the main component of the passive layer is TiO_2_ with Ti^4+^ at 459.1 eV, Ti_2_O_3_ with Ti^3+^ at 457.4 eV, TiO with at 455.1 eV, Al_2_O_3_ with Al^3+^ at 74.3 eV, and V suboxides with neglect content, respectively. It is found that the oxygen content sharply decreases with an increasing sputtering depth in the passive films (Figure 4d,h).

The passive films are covered by an adsorption oxygen layer coming from SBF solution. Under the contamination layer, TiO_2_ is detected for both samples. With an increased sputtering time, the peak site of the signal shifts toward the low-binding energy, and the peak widths of Ti 2p, Al 2p, and V 2p are broadened. It indicates the emergence of abundant suboxides beneath the outer TiO_2_ layer, Ti_2_O_3_, and TiO as the oxidation state of Ti is decreased gradually from Ti^4+^ to Ti^3+^ and Ti^2+^ mixing with Al_2_O_3_ and V suboxides. At the sputtering time of 240 s, a small amount of Ti suboxides are still discovered in the NG spectra, while they disappear for the CG sample, revealing that the NG sample is superior in depth to the CG sample. As the sputtering time is increased up to 300 s (Figure 4b,f), only small amounts of Al_2_O_3_ are detected for the NG sample, and metallic states are detected for the CG sample. It is well known that the thickness of the passive layer can be obtained based on the sputtering time of the XPS electron, which is approximately 0.5 to 1 nm for titanium alloy in the sputtering time of 10 s. Therefore, the thickness of the passive layer on the NG surface (about 20–24 nm) is much thicker than that on the CG surface (about 16 to 20 nm), which is consistent with electrochemical results.

#### 3.3.4. SEM Analysis of Passive Film

In order to characterize the composition and thickness of the passive film, SEM with EDS analysis was performed. Before the SEM observation, a nickel layer was prepared by the chemistry method and coated on the sample surface to protect the integrality of the passive layer. Figure 5a,c show the cross-sectional SEM morphologies of the formed passive film and nickel layer on the NG and CG Ti–6Al–4V samples after potentiodynamic polarization, and the corresponding composition analysis of the surface layer was analyzed by EDS (Figure 5b,d). It is obviously found that the nickel layers have been successfully coated on the NG and CG surface (Figure 5a,c), and the passive film with the distinct chromatic aberration is located in the interface between the nickel layers and alloy matrix. EDS analysis confirmed that the passive film that formed on the NG surface is much thicker than that on the CG one (Figure 5b,d). The relative content of Ti, Al, and V increased along with the scanning line from the Ti–6Al–4V matrix to the outer nickel layer, but a contrary change was observed for the Ni and O elements.

The passive film formed on both surfaces consisted of two layers with a boundary condition of the relative Ti, Al, Ni, and O content by diffraction intensity, including the outer layer (which was mainly composed of TiO_2_-rich oxide) and the inner layer of Al_2_O_3_-rich oxide. The relative content of Al_2_O_3_ in the inner passive layer of the NG sample was larger than that on the CC surface. It should be emphasized that the content of the oxidized state Al^3+^ (Al_2_O_3_) in the inner Al_2_O_3_ rich layer obtained by the above method was not accurate enough, so it was just used to assess the relative content. In this study, XPS analysis was carried out to characterize the composition and content of the passive layer, it also confirmed that the relative content of Al_2_O_3_ in the inner passive layer on the NG sample was larger than that on the CG sample (Figure 4). The above results indicate that the NG Ti–6Al–4V alloy by SFT does not change the bi-layer structure of the passive layer, but the greater thickness and the higher content of the oxidized state Al^3+^ (Al_2_O_3_) in the inner Al_2_O_3_-rich layer was obtained for the NG sample, which is the main cause for the higher compactness of the passive layer formed on the NG Ti–6Al–4V surface. 

### 3.4. Immersion

Figure 6a,c show SEM morphologies of NG and CG Ti–6Al–4V surfaces immersed in SBF solution after 240 h, and the corresponding composition of the surface layer was analyzed by the EDS method, as shown in Figure 6b,d. After immersion, it can be observed that a porous apatite layer formed on the NG sample, and a cluster of spherical-like apatite particles were distributed over the entire surface. In contrast, there is no apatite layer/particle retained on the CG surface. In addition, the EDS result reveals that the intensity of the titanium peak is suppressed by the apatite formed on the NG surface, which was attributed to the activated interaction of Ca and P over the NG surface. The formation of the apatite layer is mainly due to the higher hydrophobicity and surface energy of the NG surface, which endows the sample with an enhanced ability to attract more Ca and P ions when compared to its CG form. Therefore, with the interaction between both Ca and P in SBF, a thicker apatite layer will be formed and inevitably improve the corrosion resistance of the NG sample.

### 3.5. Mechanical Properties

Figure 7 shows the yield strength (YS), ultimate tensile strength (UTS), and fracture strain of NG and CG Ti–6Al–4V samples. It is noted that the YS and UTS increased from 882 MPa and 1020 MPa to 956 MPa and 1073 MPa, respectively, after SFT. The elongation of the NG Ti–6Al–4V alloy retained a relative higher level of 8%. In addition, the Young’s modulus was nearly unchanged after surface nanocrystallization, being 109 GPa and 112 GPa for the NG and CG samples, respectively. It is well-known that the implant materials must be strong and durable enough for bearing heavy loads and withstanding the physiological responsibilities for a long time [30]). In this study, the NG Ti–6Al–4V alloy has exhibited superior properties, including high strength, reasonable elongation, and superior corrosion resistance, which makes it a promising candidate for load-bearing implant and dental applications.

## 4. Discussion

### 4.1. Formation Mechanism of Passive Film on Both NG and CG Ti–6Al–4V Samples

The EIS, PP, and MS curves reveal that the passive layers on the NG and CG surfaces present a bi-layer structure, with an outer TiO_2_-enriched layer and a mixture inner layer of Ti oxide and Al_2_O_3_, whereas the NG sample is preponderant in the stability and compactness of passive film. During the oxidation reaction in SBF solution, the growth of passive film on Ti–6Al–4V alloy is a combine of thermodynamic and dynamic processes. Based on the passivation mechanism of Ti and Al ions containing a chloride ion electrolyte, the active sites on the sample surface are covered by an oxygen molecule layer. The transition of the ions from the metal matrix to the passive reaction interface and the formation of the passive film follow the below reactions in equations (1) to (7) [31,32]:

Anode reaction:Ti → Ti^2+^ + 2e  *E*^Θ^ = −1.630 V (298 K)(1)
Ti → Ti^3+^ + 3e  *E*^Θ^ = −1.210 V (298 K)(2)
Ti → Ti^4+^ + 4e  *E*^Θ^ = −0.730 V (298 K)(3)
Al →Al^3+^ + 3e  *E*^Θ^ = −1.660 V (298 K)(4)

Cathode reaction:Al_2_O_3_ + 3H_2_O + 3e → 2Al^3+^ + 6OH^−^  *E*^Θ^ = −1.940 V (298 K)(5)
TiO_2_ + 2H_2_O + 4e → Ti^4+^ + 4OH^−^  *E*^Θ^ = −1.270 V (298 K)(6)
H_2_O + O_2_ +4e → 4OH^−^  *E*^Θ^ = 0.401 V (298 K)(7)

Where *E*^Θ^ is the standard electrode potential of metallic ions in aqueous solution. It is found that the standard electrode potential for the transition from Al^3+^ to Al is approximately −1.660 V according to the literature [33], which is less than that for the transition from Ti^2+^ to Ti (−1.630 V), which is associated with its significantly priority oxidation ability during the passivating process. Therefore, Al_2_O_3_ starts to nucleate and stabilize after the potential exceeded −1.660 V, whereas Ti oxides will not form until a higher potential (−1.630 V) is reached. Moreover, the lower free energy of the formation for Al_2_O_3_ means that the activity of Al in the Ti–6Al–4V alloy is higher than that of the other elements during the passivation process (the free energy of formation △*G* (298 K) for TiO, TiO_2_, and Al_2_O_3_ are −495 kJ/mol, −889.5 kJ/mol, and −1582.3 kJ/mol [33]), and Al_2_O_3_ has the highest thermal stability in thermodynamics. 

Depending on the above thermodynamics analysis combined with the PP and MS results, the formation processes of passive film on both the NG and CG surfaces can be divided into three stages. At the initial stages of the standard electrode potential under –1.630 V, the passive film prefers to be Al^3+^-enriched oxide, which is stabilized by the diffusion of Al^3+^ from a metal/film interface to the film/solution interface. Meanwhile, Ti oxide is not stable, because the electrode potential is lower than −1.630 V (*E*^Θ^(Ti)). Therefore, an Al_2_O_3_-rich film is formed on both samples, while the time required for the NG sample (about 30 s) is less than that for the CG sample (about 100 s, as shown in Figure 3c). This stage is the nucleating and thickening period of the Al_2_O_3_-enriched oxide layers on both samples.

As the standard electrode potential of both NG and CG samples is in between −1.630 V and −0.730 V, Ti^2+^, Ti^3+^, Ti^4+^, and Al^3+^ diffuse to the film/electrolyte interface from the film/metal interface. However, the migration rate of Ti ions is much higher than that of Al^3+^ during the passivation process. Therefore, the passive film in this stage is a mixture of Ti oxides and Al_2_O_3_, with Ti oxides being the principal component. At the end of the oxide reaction, a topmost surface layer constituted completely by TiO_2_ will form. The schematic diagrams for the passive process of the Ti–6Al–4V samples are illustrated in Figure 2f.

### 4.2. Effect of NG Microstructure on Corrosion Behavior

The results of the OCP, PDP, and EIS tests reveal that the nanostructured Ti–6Al–4V alloy can rapidly form a more stable and protective oxide film in SBF solution, resulting in its significantly improved corrosion performance, which is also in accordance with the immersion test (Figure 6), PP (Figure 3c), and MS analysis (Figure 3d). In addition, XPS analysis shows that the composition of passive layers on the NG and CG samples is very similar. It is well known that the nanostructure is characterized by its high density of grain boundaries, which is beneficial for the nucleation and growing of a homogeneous and dense passive film [34], leading to the thickened passive film in the NG sample. It also contributes greatly to the superior electrochemical corrosion properties of the NG sample relative to its CG form.

### 4.3. Effect of NG Structure on Mechanical Properties

The gradient nanostructed Ti–6Al–4V exhibits improved strength and significantly reduced ductility, which can be attributed to its unique microstructure. The enhanced strength comes from the grain refinement in the surface layer and the unique heterogeneous structure. It has been verified that an additional strength can be achieved in several gradient nanograined materials due to the strain gradient induced between the NG layer and underneath the CG core [35,36]. In addition, the intrinsic tensile plasticity of the NG surface layer can be inspired as the strain localization is effectively suppressed by the CG core [37]. So, a moderate elongation can be obtained, which correlated well with the fracture morphologies in Figure 7b–e. The NG Ti–6Al–4V sample presents a large number of dimples and a few cleavage planes in the central crack propagation region (Figure 7b,c), and the edge shear lip region consists of a mixture of shallow and parabolic dimples and shear planes. This appearance of both coarse/parabolic dimples and smooth shear planes indicates the relative poor ductility, which belongs to the mixed fracture of toughness and brittleness. The CG Ti–6Al–4V sample (Figure 7d,e) presents the ductile fracture mode with a large number of dimples in both the propagation and the shear-lip region. Therefore, the NG Ti–6Al–4V alloy possesses an excellent combination of high strength, low modulus, and reasonable ductility for many load-bearing implant and dental applications.

## 5. Conclusions

(1) The NG surface layer with an average grain size of ≤100 nm is successfully fabricated on a Ti–6Al–4V sheet by means of SFT.

(2) The NG sample possesses a higher *E*_corr_, *R*_p_ + *R*_b_ and lower *I*_pp_, which indicates the higher corrosion resistance. 

(3) The passive films for the NG and CG samples have the similar bi-layer structure, including a TiO_2_-enriched outer layer and a mixture of Ti oxide and an Al_2_O_3_ in the inner layer. 

(4) The improved corrosion performances of the NG sample can be attributed to its enhanced thickness and compactness compared with the CG sample, which is associated with its unique microstructure.

(5) The NG Ti–6Al–4V sample exhibits a significantly enhanced strength and a moderate ductility.

## Figures and Tables

**Figure 1 materials-12-00760-f001:**
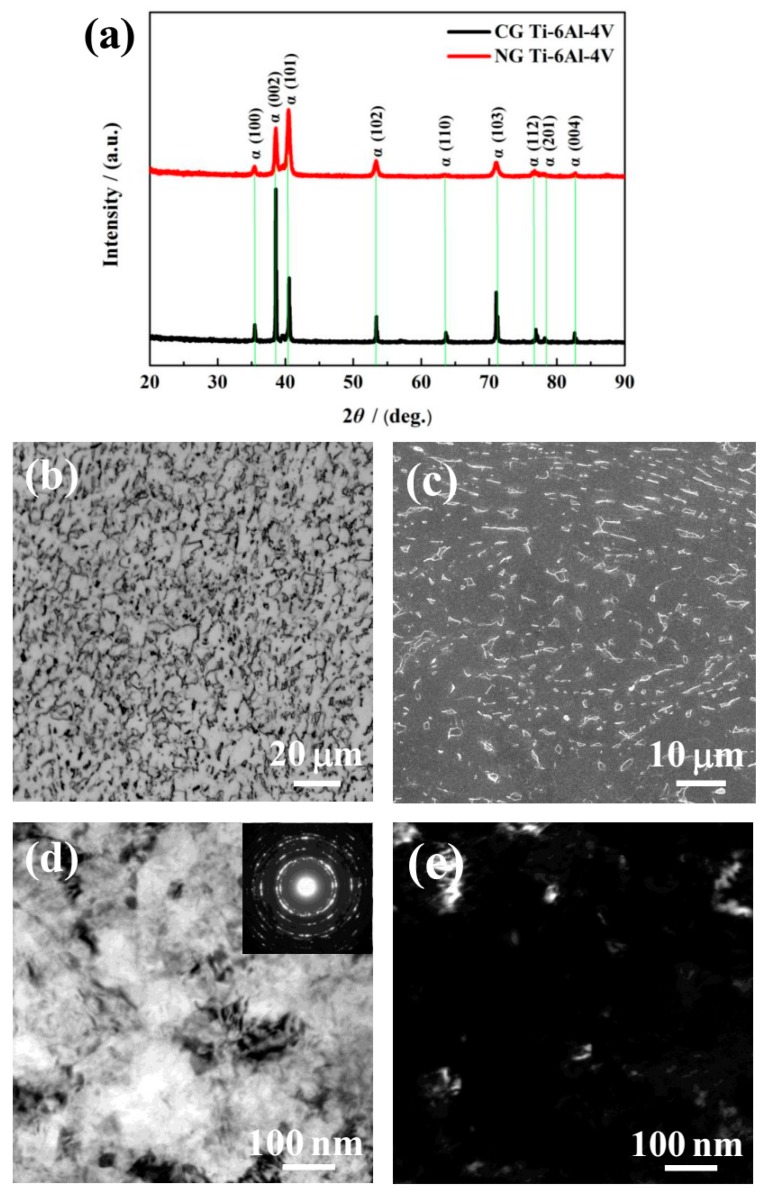
X-ray diffraction (XRD) patterns of nanograined (NG) and coarse-grained (CG) Ti–6Al–4V alloys (**a**). Optical micrograph (**b**) and SEM magnified images (**c**) for CG microstructure, TEM bright-field (**d**), and dark-field (**e**) micrograph for NG microstructure.

**Figure 2 materials-12-00760-f002:**
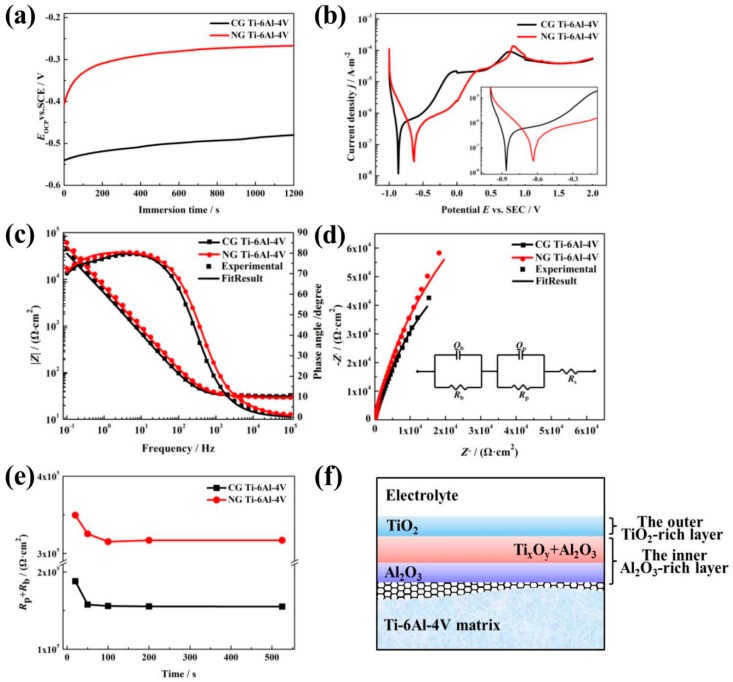
Open circuit potential (OCP) plots (**a**), PDP curves (**b**), Bode plots (**c**), and the corresponding Nyquist plots (**d**) of NG and CG Ti–6Al–4V samples in SBF solutions at 37 °C. Variation of *R*_p_ + *R*_b_ and testing time in applying passive potential (**e**), and equivalent electrical circuit (insert in (**d**)) with *R*_s_(*Q*_p_*R*_p_)(*Q*_b_*R*_b_) model used to fit the impedance spectra. Schematic diagrams (**f**) of passive layers formed on NG and CG Ti–6Al–4V surfaces.

**Figure 3 materials-12-00760-f003:**
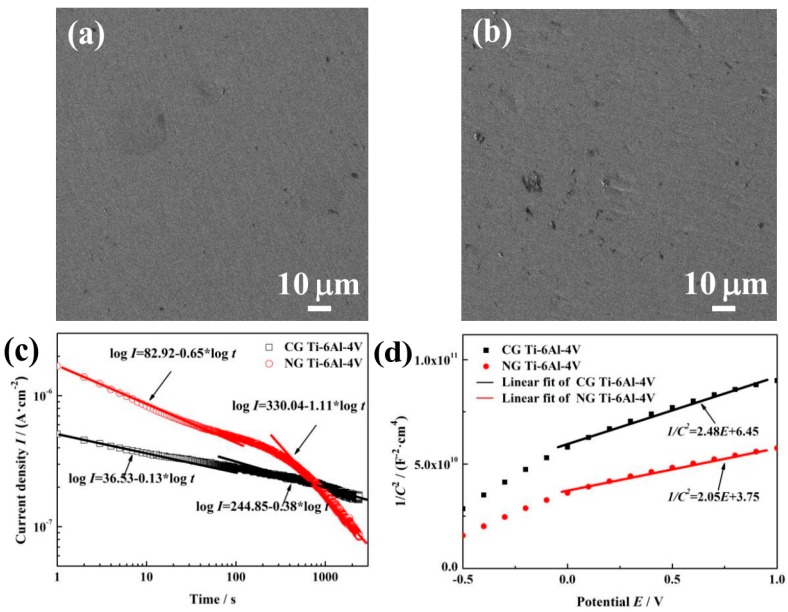
SEM surface micrographs of NG (**a**) and CG (**b**) samples after potentiodynamic polarization. Current vs. time curves (**c**) and Mott–Schottky curves (**d**) for NG and CG samples in SBF solutions at 37 °C.

**Figure 4 materials-12-00760-f004:**
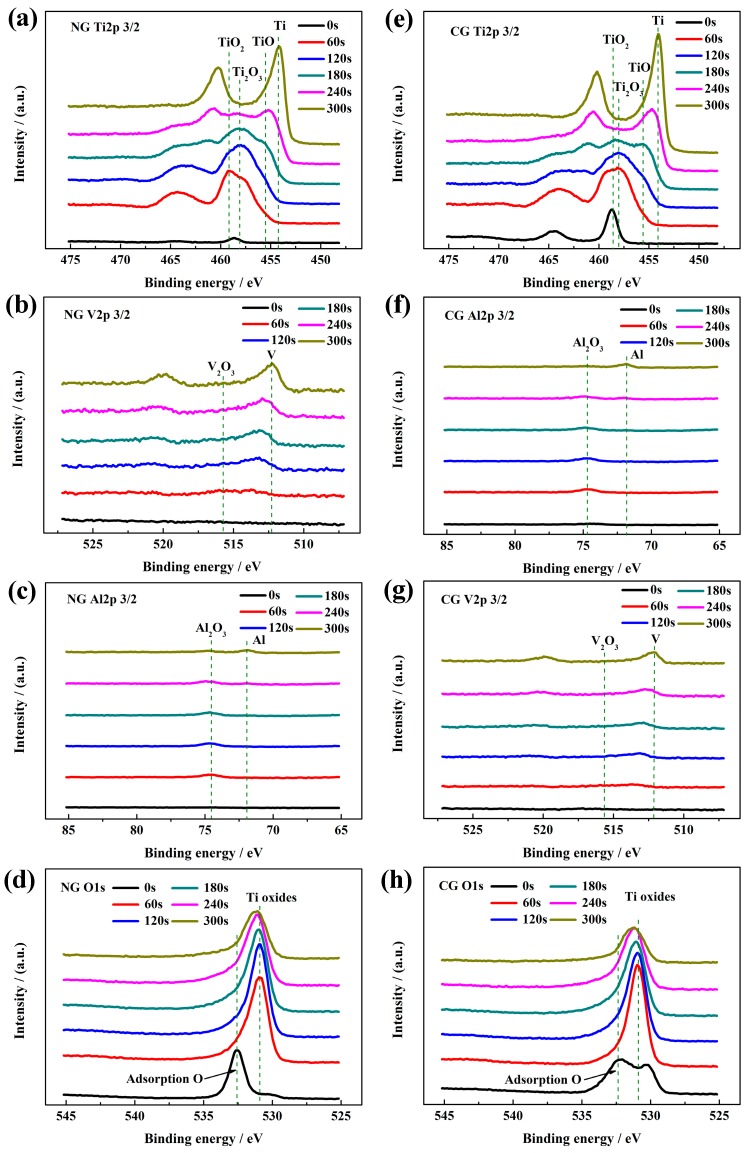
High-resolution XPS spectra of NG and CG Ti–6Al–4V alloys after polarization with different sputtering depths: (**a** and **e**) Ti 2p, (**b** and **f**) Al 2p, (**c** and **g**) V 2p, and (**d** and **h**) O 1s.

**Figure 5 materials-12-00760-f005:**
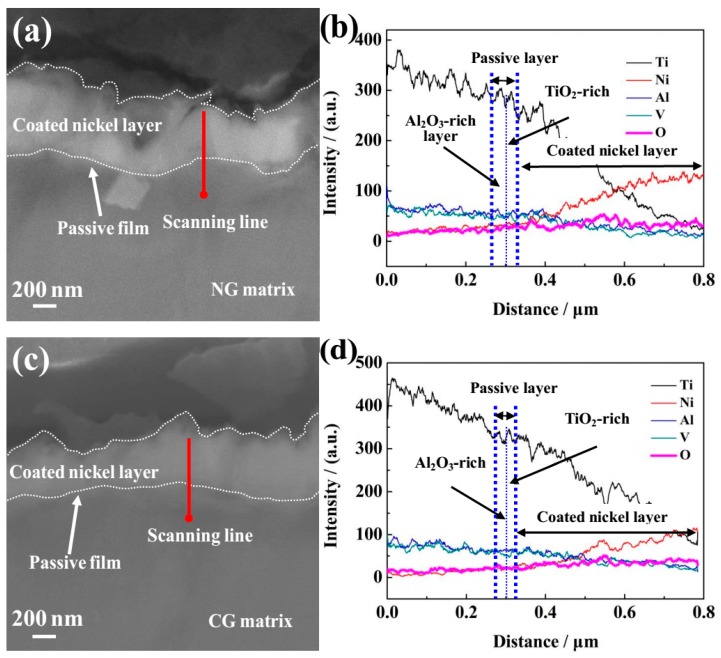
Cross-sectional SEM observations of passive films formed on NG (a) and CG (c) Ti–6Al–4V alloys after potentiodynamic polarization. The corresponding energy-dispersive spectrometer (EDS) analyses for NG (**b**) and CG (**d**) alloys correspond to red lines are shown in (**a**) and (**c**).

**Figure 6 materials-12-00760-f006:**
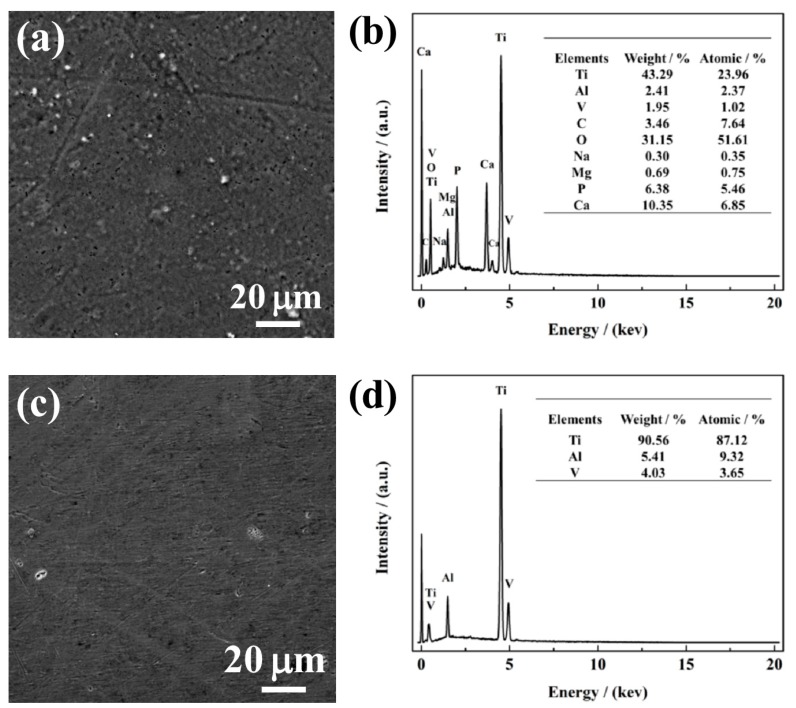
SEM micrographs with the corresponding EDS analysis of NG (**a** and **b**) and CG (**c** and **d**) Ti–6Al–4V alloys immersed in SBF solution for 720 h.

**Figure 7 materials-12-00760-f007:**
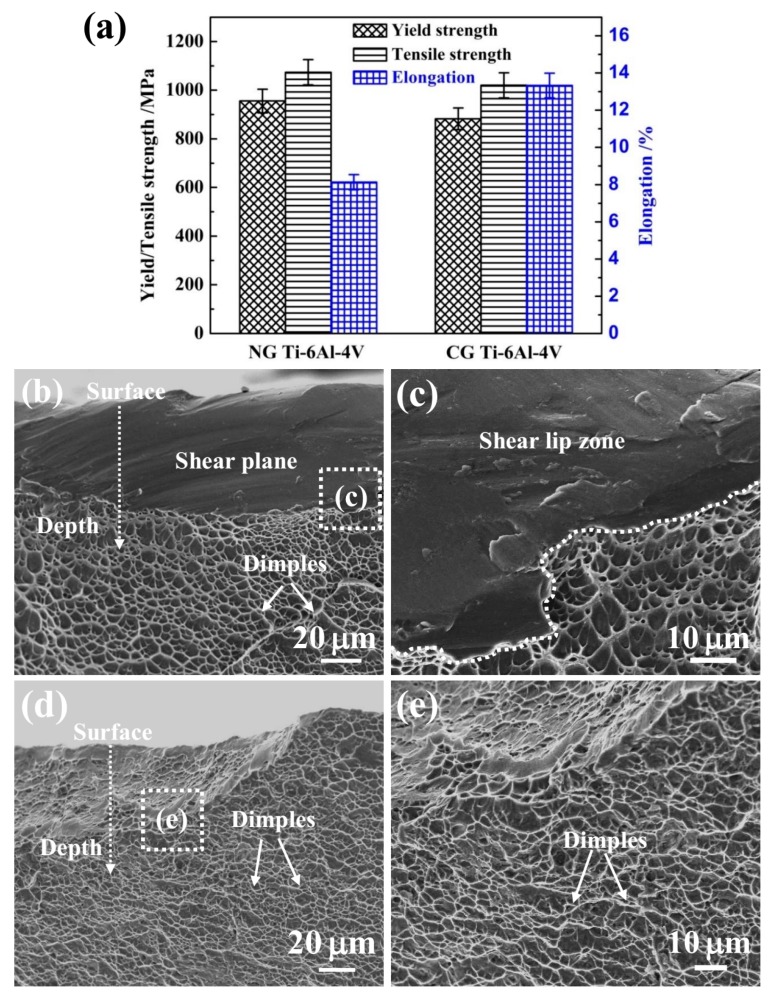
Mechanical properties (**a**) of NG and CG Ti–6Al–4V alloys. SEM images of fracture surface of shear lip zone for NG (**b**) and CG (**d**) Ti–6Al–4V alloy in the edge region, (**c**) and (**e**) are high-magnified fracture morphologies for the zone in Figure 7**b** and Figure 7**b**.

**Table 1 materials-12-00760-t001:** Corrosion parameters of potentiodynamic polarization (PDP) for CG and NG Ti–6Al–4V alloys in simulated body fluid (SBF) solutions at 37 °C.

Samples	*E*_corr/_V	*I*_pp/_μA/cm^2^
CG	–0.816±0.023	26.280±0.150
NG	–0.641±0.015	25.387±0.103

**Table 2 materials-12-00760-t002:** Electrochemical impedance parameters of the CG and NG Ti–6Al–4V alloys in SBF solution at 37 °C.

Samples	*R*_s_ /Ω·cm^2^	*R*_p_ /kΩ·cm^2^	*Q*_p_ /F/cm^2^	*n* _1_	*R*_b_ /kΩ·cm^2^	*Q*_b_ /μF/cm^2^	*n* _2_	Chi-square
CG	31.818 ± 0.072	2.714 ± 0.356	98.980 ± 0.061	0.826 ± 0.081	150.002 ± 0.261	38.678 ± 0.090	0.922 ± 0.072	0.000502
NG	29.371 ± 0.048	5.175 ± 0.120	83.959 ± 0.281	0.802 ± 0.054	314.203 ± 0.179	30.546 ± 0.103	0.906 ± 0.060	0.000235

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
