# Peer review of "Electrochemical Corrosion Behavior and Mechanical Properties of Nanocrystalline Ti–6Al–4V Alloy Induced by Sliding Friction Treatment"

_materials, 2019, doi:10.3390/ma12050760_

Round 1

Reviewer 1 Report

The authors have investigated the interesting and important topic, but the text itself causes a number of negative comments:
- The title is not clear.
-  The abbreviation CG in abstract is not explained. It will be better to avoid the abbreviations in abstract at all.
- The explanation of passive layer preparation is missing.
- What lattice distortion do the authors mean, explaining the fig. 1a? Why the formation of NG layer should result in the lattice distortion?
- Scale in fig. 1b-f is missing. From fig. 1e,f it is impossible to make a conclusion about the presence of grains, which are less than 100 nm. The inset of fig. 1d is missing.
- The order of a,b,c,d parts of fig.4 is missing.
- The evidence of oxide passive layers existence appears only in fig. 5, but the discussion about it starts in fig. 2.
- The fig. 4 has not enough resolution to see any oxide layer in it. It is absolutely unclear the positions of borders of passive layers (vertical dashed lines) and especially the thicknesses of them in fig. 4 c,d. Why is the thickness of layer on NG sample thicker?
- What about the vanadium oxides? Why the authors do not take them into the consideration?
- What does it mean: "typical and enlarged spectra" in fig. 5? Where is the typical and where is the enlarged one?
- Conclution in lines 379-381 is not clear. What is exactly happend in fig. 3c at 30 and 100 s.

Author Response

Reviewer 1#:

1). The title is not clear.

Response:

We have checked the title very carefully, and find that the abbreviation of “SFT” is not clear for the reader. So the title has been revised as “Electrochemical corrosion and mechanical properties of nanocrystalline Ti-6Al-4V alloys induced by sliding friction treatment”.

2). The abbreviation CG in abstract is not explained. It will be better to avoid the abbreviations in abstract at all.

Response:

The full name of CG has been added in the abstract, and we are also checked the whole manuscript to avoid the similar errors.

3). The explanation of passive layer preparation is missing.

Response:

The passive layer study in this manuscript is the electrolyte-formed oxides on the sample surface, which is began to form after the electrode immersing into the electrolyte, and then kept in solutions until a stable corrosion potential is reached. Meanwhile, the stabilized passive layer is also obtained in this process. This supplement information is added and marked in red at the section of “2.3 Electrochemical measurements” in the manuscript.

4). What lattice distortion do the authors mean, explaining the Fig. 1a? Why the formation of NG layer should result in the lattice distortion?

Response:

Thanks for the reviewer’s comments. During plastic deformation, the dislocation sources (Frank-Read sources) become activated, and a large number of dislocations and vacancies occur in the surface deformation layer. With the continuous increase of the strain and strain rate by SFT, the dislocation strain fields can interact with each other, which makes some atoms in the crystal deviate from the original equilibrium position, resulting in lattice distortion and grain refinement. In XRD patterns (Fig. 1(a)), the main diffraction peaks of SFT-treated (NG) sample are considerably broadened. Therefore, the lattice distortion and different microstructures featured of dislocation tangles, dislocation walls and lamella shaped subgrains are mainly closer to the surface NG layer. This supplement information is added and marked in red at the section of “3.1 Phase constitutions and microstructure” in the manuscript.

5). Scale in Fig. 1b-f is missing. From fig. 1e,f, it is impossible to make a conclusion about the presence of grains, which are less than 100 nm. The inset of fig. 1d is missing.
Response:

Thanks for pointing out the mistake. The scale in Fig. 1(b-f) and SAED pattern inset Fig. 1(d) have been added. As shown in TEM images of NG microstructure in Fig. 1(d, e), the NG sample consists of α grains of average size smaller than 100 nm in the top surface layer, the SAED pattern (inset Fig. 1(d)) with a discontinuous diffused diffraction rings also confirms that the grains have been refined into nano-scale with grain boundaries in non-equilibrium state.

6). The order of a,b,c,d parts of fig.4 is missing.

Response:

Thanks for pointing out the mistake. The order of a,b,c,d parts in Fig.4 have been added.

7). The evidence of oxide passive layers existence appears only in Fig. 5, but the discussion about it starts in Fig. 2.

Response:

The passive layer study in this manuscript is the electrolyte-formed oxides on the sample surface, which is began to form after the electrode immersing into the electrolyte, and then kept in solutions until a stable passive layer is obtained. Fig. 2 shows the electrochemical corrosion results of the passive film formed on the surface of NG and CG samples, and Fig. 5 shows the typical and enlarged XPS spectra of NG and CG Ti-6Al-4V alloys after polarization with different sputtering depths. So the analysis and discussion of the passive layers is started from Fig. 2.

8). The Fig. 4 has not enough resolution to see any oxide layer in it. It is absolutely unclear the positions of borders of passive layers (vertical dashed lines) and especially the thicknesses of them in Fig. 4 c,d. Why is the thickness of layer on NG sample thicker?

Response:

Thanks for the reviewer’s comments. It is as reviewer suggested that there is no obviously layer boundary for passive film. The unclear borders between passive film and matrix are mainly attributed to in situ oxidation reaction on the sample surface, and the smaller color contrast and the thinner passive layer make it difficult to distinguish in SEM images. In this study, the thickness of passive film can be quantitatively measured by Energy Dispersive Spectrometer (EDS) and X-ray photoelectron spectrometer (XPS). EDS analysis and XPS spectra are confirmed that the passive film on the NG alloy is a much thicker than that on the CG alloy.

9). What about the vanadium oxides? Why the authors do not take them into the consideration?

Response:

In generally, the composition of passive film is constituted by the mixture of TiO2, Al2O3 and V oxides, but the diffraction peak of V oxides in XPS spectra for passive films is vanished due to the much higher standard electrode potential (EΘ) for V oxides (-1.190 V for VO, -0.260 V for V2O3, 0.360 V for VO2 and 1.0 V for VO) relative to the values of EΘ for TiO2 (-1.630) and Al2O3 (-1.660). Accordingly, a significantly higher standard electrode potential is needed to form V oxides in the electrolyte solution. Therefore, V suboxides/oxides have not been discovered in the XPS spectra of passive films.

10). What does it mean: "typical and enlarged spectra" in Fig. 5? Where is the typical and where is the enlarged one?

Response:

Thanks for the reviewer’s comments. We have made the corresponding modification for the term of “typical and enlarged spectra”, and which is revised as “high-resolution”.

11). Conclution in lines 379-381 is not clear. What is exactly happend in Fig. 3c at 30 and 100 s.

Response:

Thanks for the reviewer’s comments. In this study, the potentiostatic polarization (PP) curve with the variation of current density (I) and time (t) can be used to reflect passive film compactness, the double-log plots for NG and CG samples are shown in Fig. 3(c). For NG and CG samples, the current density with time at 30 and 100s is the critical value for the passive film from porosity to density.

Reviewer 2 Report

In this manuscript, a nano-grained surface layer has been fabricated on Ti-6Al-4V by a sliding friction treatment (SFT) to improve the corrosion and mechanical performance. The following problems should be addressed before publication:

Abbreviations should be avoided in the title.

In Fig.1 and Fig. 3, there is no scale bar.

In Fig.2, there is just one time-constant appeared in the EIS patterns. This is not consistent with the two time-constants in the equivalent electrical circuit and the three layers in the schematic diagrams.

Ti alloys have high corrosion resistance and good biocompatibility, and the corroded surface in Fig. 6a showed a good inductivity for the CaP precipitation, while the NG Ti surface didn't, indicating a decreased surface bioactivity. How do the authors think of the effects of the surface treatment on its biocompatibility?

Author Response

Reviewer 2#:

1). In Fig.1 and Fig. 3, there is no scale bar.

Response:

Thanks for pointing out the mistake. The scale bars in Fig. 1 and Fig. 3 have been added.

2). In Fig.2, there is just one time-constant appeared in the EIS patterns. This is not consistent with the two time-constants in the equivalent electrical circuit and the three layers in the schematic diagrams.

Response:

It is true as Reviewer said that the equivalent electron circuit with a single time constant can be obtained from the Fig. 2(c), and this is mainly attributed to the narrower frequency region (10-1 - 105), which leads to the missing of the other time constant in Bode plots. This phenomenon is also observed by Wei et al. (H Wei, Y Wei, L Hou, N Dang, Correlation of ageing precipitates with the corrosion behavior of Cu-4wt.% Ti alloys in 3.5wt.% NaCl solution, Corrosion Science. 111 (2016) 382-390.) and Wang et al. (Z.B. Wang, H.X. Hua, Y.G. Zheng, W. Kea, Y.X. Qiao, Comparison of the corrosion behavior of pure titanium and its alloys in fluoride-containing sulfuric acid, Corrosion Science. 103 (2016) 50-65.). The EIS analysis by Wei is shown as follow:

Therefore, the model analysis of EIS data using an equivalent circuit with two time constants is reasonable in this study. Moreover, the fitting quality of the equivalent electron circuit with two time constants is evaluated by the quite small chi-square values of about 10−4, as shown in Table 2. It well fits the result in this study.

3). Ti alloys have high corrosion resistance and good biocompatibility, and the corroded surface in Fig. 6a showed a good inductivity for the CaP precipitation, while the NG Ti surface didn't, indicating a decreased surface bioactivity. How do the authors think of the effects of the surface treatment on its biocompatibility?

Response:

Thanks for the reviewer’s comments. In this study, Fig. 6(a) and Fig. 6(c) show SEM surface morphologies of NG and CG Ti-6Al-4V samples immersed in SBF solution after 240 h, and the corresponding composition of the surface layer was analyzed by EDS method, as shown in Fig. 6(b) and Fig. 6(d). After immersion, it can be observed that a porous apatite layer has formed on NG sample, and a cluster of spherical like apatite particles are distributed over the entire surface. The EDS result reveals that the intensity of titanium peak is suppressed by apatite formed on NG surface, attributed to the activated interaction of Ca and P over the NG surface. Whereas, there is no apatite layer/particle retained on CG surface. Therefore, the NG sample exhibits the high corrosion resistance and good biocompatibility.

Moreover, the electrochemical corrosion characteristics and biocompatibility of nanostructured titanium for implants have been investigated by the author, which is published in the journal of “Applied Surface Science” (Applied Surface Science 434 (2018) 63–72). The result showed that the NG sample exhibited an excellent combination of in vitro biological anti-corrosion and biocompatibility properties. The better corrosion resistance for NG sample is attributed to the formation of a thicker passive film, which delays the electron migration and electrochemical reactions between nanocrystalline surface and ion transfer. The improved osteoblastic cells response on NG sample is due to the higher wettability, surface energy and nano-roughness caused by the presence of nanocrystal structures.

Reviewer 3 Report

The paper concerns an interesting and comprehensive study of surface nanostructured Ti-6Al-4V alloy, using principally advanced electrochemical techniques, electron microscopy, XPS and mechanical testing. The manuscript is well organized and in the Journal scope. Suggested revisions and comments are following:

1.      In the abstract: “Tensile experiments … to evaluate elasticity properties at room temperature.” The term “elasticity properties” is not suitable since the full set of mechanical properties under tension were determined. Please replace this term with “mechanical properties”.

2.      Scale markers in the micrographs of Figures 1 and 3 are not visible. Please check and advise.

3.      In general, some Figures were separated in two pages. Please organize the Figures, in such way, in order to be contained in one single page. Also in Figure 2, “(e) sign” of the corresponding sub-figure is missing.

4.      Please keep one decimal digit, as the significant one, for EDS quantitative results, listed in Figure 6.

5.      Figure 7 needs enlargement to enhance more details. In the same Figure, in the caption, it should be written (b) and (c), instead of (a) and (b). The (a) is written twice. The (c) is missing.

6.      The fracture mechanisms depicted by the respective fractographs, need to be addressed in more detail in paragraph 3.5 (Fig. 7b and c). Dimpled fracture is almost common feature in both materials (NG and CG alloys), but “cleavage” fracture is not readily evidenced. In the last lines of discussion section, the comments for fracture mechanisms are conflicting, compared with what was described in the relevant caption (Fig. 7). Please check and advise.

7.      The format and style of reactions and the electrochemical potential values (section 4.1) need editorial improvement, in order to be depicted in a more well-appeared and clearly written fashion. The same comment is made for the equation in line 231 too.

8.      The ductility seems that it was significantly reduced, elongation decrease from ~13% to ~8%. Please amend and comment the above statement, in relation to lines 401-402.

9.      The corrosion current density is presented in terms of Jpp (in Table 1 and section 3.2.2), while in the conclusions section the Icorr is referred (L414). Also, a different value of current density is written for CG alloy in Table 1, compared with that referred in Line 172. Please check and advise.

10.   The language seems sufficient; however the correction of minor spelling errors and editorial discrepancies during a final proofreading is necessary. For example:

-        In the abstract please write the stoichiometric expressions of chemical compounds using the formal style: using the subscripts to indicate the number of atoms per unit cell (L13).  

-        Orthopaedic or orthopedic: please adopt a uniform writing style (L24 vs. L28).

-        L236: there are (there is).

Author Response

Reviewer 3#:

1). In the abstract: “Tensile experiments … to evaluate elasticity properties at room temperature.” The term “elasticity properties” is not suitable since the full set of mechanical properties under tension were determined. Please replace this term with “mechanical properties”.

Response:

Thanks for the reviewer’s suggestion. The term of “elasticity properties” is revised as “mechanical properties”.

2). Scale markers in the micrographs of Figures 1 and 3 are not visible. Please check and advise.

Response:

Thanks for pointing out the mistake. The scale bars in Fig. 1 and Fig. 3 have been added.

3). In general, some Figures were separated in two pages. Please organize the Figures, in such way, in order to be contained in one single page. Also in Figure 2, “(e) sign” of the corresponding sub-figure is missing.

Response:

Thanks for pointing out the mistake. We have checked all the Figures very carefully to avoid the wrong figure distribution, and the number of “(e)” for Fig. 2 also has been added.

4). Please keep one decimal digit, as the significant one, for EDS quantitative results, listed in Figure 6.

Response:

Thanks for the reviewer’s comments. Energy Dispersive Spectrometer (EDS), is an analytical technique used for the elemental analysis or chemical characterization of a sample. Its characterization capabilities are due in large part to the fundamental principle that each element has a unique atomic structure allowing a unique set of peaks on its electromagnetic emission spectrum. In this study, two digits after the decimal point can increase efficiency and the calculation precision for the chemical composition of the corroded surface. This similar expression is also used by Liu et al (Electrochim Acta, 233 (2017) 190-200).

5). Figure 7 needs enlargement to enhance more details. In the same Figure, in the caption, it should be written (b) and (c), instead of (a) and (b). The (a) is written twice. The (c) is missing.

Response:

As reviewer suggested that the high-magnification SEM image has been provided in the Fig. 7, and the correct caption is also added.

6). The fracture mechanisms depicted by the respective fractographs, need to be addressed in more detail in paragraph 3.5 (Fig. 7b and c). Dimpled fracture is almost common feature in both materials (NG and CG alloys), but “cleavage” fracture is not readily evidenced. In the last lines of discussion section, the comments for fracture mechanisms are conflicting, compared with what was described in the relevant caption (Fig. 7). Please check and advise.

Response:

Thanks for the reviewer’s suggestion and we have made deep analysis and discussion on fracture mechanisms. The revised content is as follows

So a moderate elongation can be obtained, correlated well with the fracture morphologies in Fig. 7(b-e). NG Ti-6Al-4V sample exhibits a large number of dimples and a few cleavage fracture planes in the central crack propagation region (Fig. 7(b) and Fig. 7(c)), and the edge shear lip region consists of the mixture of shallow and parabolic dimples and shear planes. This appearance of both coarse/parabolic dimples and smooth shear planes indicate the relative poor ductility, which belongs to the mixed fracture of toughness and brittleness. And the fracture characteristic of CG Ti-6Al-4V sample is absolutely different. CG Ti-6Al-4V sample (Fig. 7(d) and Fig. 7(e)) shows a ductile fracture mode with plenty of dimples and cleavage fracture planes in both of propagation and shear-lip region. Therefore, NG Ti-6Al-4V alloy has achieved an excellent combination of high strength, low modulus, reasonably ductility properties for many load-bearing implant and dental applications.

7). The format and style of reactions and the electrochemical potential values (section 4.1) need editorial improvement, in order to be depicted in a more well-appeared and clearly written fashion. The same comment is made for the equation in line 231 too.

Response:

Thanks for the reviewer’s suggestion, and we will cooperate with the editor to further improve the format and style of reactions and the electrochemical potential values.

8). The ductility seems that it was significantly reduced, elongation decrease from ~13% to ~8%. Please amend and comment the above statement, in relation to lines 401-402.

Response:

It is true as Reviewer said that the ductility of NG sample exhibits a significantly reducing, and the elongation is decreased form 13% (CG sample) to 8% (NG sample). We have made the corresponding revision as “The gradient nanostructed Ti-6Al-4V exhibits improved strength and significantly reduced ductility, which can be attributed to its unique microstructure.” (lines 404-405).

9). The corrosion current density is presented in terms of Jpp (in Table 1 and section 3.2.2), while in the conclusions section the Icorr is referred (L414). Also, a different value of current density is written for CG alloy in Table 1, compared with that referred in Line 172. Please check and advise.

Response:

Thanks for pointing out the mistake. The passive current density is all uniformly expressed as “jpp”. And the same passive current density of CG sample (26.280 μA/cm2) is added in the manuscript and Table 1.

10). The language seems sufficient; however the correction of minor spelling errors and editorial discrepancies during a final proofreading is necessary. For example:

-        In the abstract please write the stoichiometric expressions of chemical compounds using the formal style: using the subscripts to indicate the number of atoms per unit cell (L13).  

-        Orthopaedic or orthopedic: please adopt a uniform writing style (L24 vs. L28).

-        L236: there are (there is).

Response:

Thanks for pointing out the mistake. We have checked the whole manuscript very carefully, and some spelling/grammatical errors and badly chosen terms have been revised. Here, we did not list the revised portion but marked in red in the revised manuscript.

Round 2

Reviewer 1 Report

- Please, explain carefully, how did you estimate the thickness of the passive layer. It is impossible to make it from the data of fig.4, but you are talking about the thickness in comments to it (line 279-280). Explain, why the EDS analysis confirmed that the passive film on the NG alloy is a much thicker than that on the CG alloy (line 276). Do you see it in fig.4 (c) and (d)? Are there some changes of element concentrations near the 0.3 μm to confirm it? But there is nothing like this in fig.4.  I see only the drawn vertical lines in different positions without explanations.
- Where do you see "the much higher content of oxidized state Al3+ (Al2O3) in the inner Al2O3 rich layer" in fig.4 (line 287)?
- Obviously, XPS analysis can estimate the thickness of the passive layer. So, please, do it, connecting the sputtering time and the thickness.
- On my opinion the XPS peaks concerned to Al2O3 are the same for sputtering time from 60 to 240 seconds (fig.5 (b)). If you realy see "the inner Al2O3 rich layer" here, please, explain it. By the way, may be if you change the places of parts 3.3.3 and 3.3.4 in the text, it will help to understand it.
- The parts (a)-(d) of fig.4 are mentioned in wrong order (again) both in text (lines 274 and 277) and in figure caption.
- The fig.5 caption contains "the typical and enlarged" spectra (again).
- The table 2 is unreadable.
- What does it mean "CC sample" in line 284?

Author Response

Reviewer #1:

1). Please, explain carefully, how did you estimate the thickness of the passive layer. It is impossible to make it from the data of fig.4, but you are talking about the thickness in comments to it (line 279-280). Explain, why the EDS analysis confirmed that the passive film on the NG alloy is a much thicker than that on the CG alloy (line 276). Do you see it in fig.4 (c) and (d)? Are there some changes of element concentrations near the 0.3 μm to confirm it? But there is nothing like this in fig.4.  I see only the drawn vertical lines in different positions without explanations.

Response: Thanks for the reviewer’s suggestion. In this study, the thickness of the passive layer is mainly estimated by using XPS spectra and SEM with EDS analysis. It is true as Reviewer said that the thickness of passive layer is not quantitatively confirmed by SEM with EDS analysis, and which is only a qualitative analysis. The relative thickness of the passive layer is estimated by the content evolutions of Al, O and Ti along the drawn vertical lines (the scanning line) from the matrix to the surface, as shown in Fig. 5(b) and Fig. 5(d). The relative content of Ti, Al and V increases with along the scanning line from Ti-6Al-4V matrix to the outer nickel layer, but the contrary change is observed for Ni and O elements. In addition, it should be emphasized that relative thickness of the passive layer obtained by the method above is not accurate enough, so it is just used to assess the relative thickness. In this study, XPS analysis is performed to characterize the composition and thickness of the passive film, it is also confirms that the passive film on NG sample is much thicker than that on CG sample. It is well known that the thickness of passive film can be obtained based on the sputtering time of XPS electron, which is approximately 1 ~ 3 nm for titanium alloy in the sputtering time of 10 s. Therefore, the thickness of passive layer on NG surface (about 20 ~ 24 nm) is much thicker than that on CG surface (about 16 ~ 20 nm), which is consisted with electrochemical results.

Generally speaking, the thickness of passive layer on titanium alloys is smaller than 50 nm, which is usually characterized by TEM with EDS analysis. The film thickness of Ti-0.2Pd was 22 nm in H2SO4 solutions (pH=1) with 0.003 M fluoride ions after 24 h immersion, and the thickness for Ti-0.3Mo-0.8Ni was less than 5 nm (Corrosion Science 103 (2016) 50-65); The passive film layer for nanostructured Ti2448 alloy in 0.9% NaCl solution at 37 °C was 18 nm, a little thicker than that on its coarse-grained form of 13 nm (Acta Biomaterialia 10 (2014) 2866 - 2875). Moreover, the thickness of passive film can be obtained by the calculations based on the electrochemical result (Materials Science and Engineering C 71 (2017) 771-779), and which is also an important work for us in future.

2). Where do you see "the much higher content of oxidized state Al3+ (Al2O3) in the inner Al2O3 rich layer" in Fig.4 (line 287)?

Response: Thanks for the reviewer’s comments. The relative content of oxidized state Al3+ (Al2O3) in the inner Al2O3 rich layer is estimated by the evolutions of Al, O and Ti over the distance from the surface, as shown in Fig. 5(b) and Fig. 5(d). It should be emphasized that the content of oxidized state Al3+ (Al2O3) in the inner Al2O3 rich layer obtained by the method above is not accurate enough, so it is just used to assess the relative content. In this study, XPS analysis is performed to characterize the composition and content of the passive film, it is also confirms that the relative content of Al2O3 in the inner layer of passive film on NG sample is larger than that in the CC sample (Fig. 4).

This supplement information is added and marked in red at the section of “3.3.4. SEM analysis of passive film” in the manuscript.

3). Obviously, XPS analysis can estimate the thickness of the passive layer. So, please, do it, connecting the sputtering time and the thickness.

Response: Thanks for the reviewer’s suggestion. It is true as Reviewer said that the thickness of passive film can be obtained based on the sputtering time of XPS electron (approximately 0.5-1 nm for titanium alloy in the sputtering time of 10s). Therefore, the thickness of passive layer on NG surface (about 20 ~ 24 nm) is much thicker than that on CG surface (about 16 ~ 20 nm), which is consisted with electrochemical results. This supplement information is added and marked in red at the section of “3.3.3. XPS analysis of passive film” in the manuscript.

4). On my opinion the XPS peaks concerned to Al2O3 are the same for sputtering time from 60 to 240 seconds (fig.5 (b)). If you realy see "the inner Al2O3 rich layer" here, please, explain it. By the way, may be if you change the places of parts 3.3.3 and 3.3.4 in the text, it will help to understand it.

Response: It is true as Reviewer said the XPS peaks concerned to Al2O3 for NG and CG samples are the same for sputtering time from 60 to 240 seconds (Fig.4 (b) and Fig.4 (f)). But as the sputtering time is increased up to 300 s, small amounts of Al2O3 is detected for NG sample, and only metallic states are detected for CG sample, indicates that the contents of Al2O3 is higher than that in CG one. In addition, the order of parts 3.3.3 and 3.3.4 in the text has been changed as the reviewer’s suggestion.

5). The parts (a)-(d) of fig.4 are mentioned in wrong order (again) both in text (lines 274 and 277) and in figure caption.

Response: Thanks for pointing out the mistake. The right order for the parts (a)-(d) in Fig.4 has been revised in test and figure caption.

6). The fig.5 caption contains "the typical and enlarged" spectra (again).

Response: Thanks for pointing out the mistake. “Typical and enlarged XPS spectra” has revised as “High-resolution XPS spectra” in Fig. 5 caption.

7). The table 2 is unreadable.

Response: Thanks for pointing out the mistake. We have made the corresponding correction for the format of Table 2.

8). What does it mean "CC sample" in line 284?

Response: Thanks for pointing out the mistake. “CC sample” has revised as “CG sample” in the manuscript (line 284).

Reviewer 2 Report

Accept in present form.

Author Response

Reviewer #2:

1): [Accept].

Response: Thanks for the reviewer’s approval.

Reviewer 3 Report

The paper could be accepted, as it was amended. As a further recommendation, the phrase "And the fracture characteristic of CG Ti-6Al-4V sample is absolutely different." (L416-417) is quite excessive comparing the fracture mechanisms of the two samples. It is proposed to be ommited during proof reading.

Author Response

Reviewer #3:

1): The paper could be accepted, as it was amended. As a further recommendation, the phase "And the fracture characteristic of CG Ti-6Al-4V sample is absolutely different." (L416-417) is quite excessive comparing the fracture mechanisms of the two samples. It is proposed to be ommited during proof reading.

Response: Thanks for the reviewer’s approval. The sentence of “And the fracture characteristic of CG Ti-6Al-4V sample is absolutely different.” has been deleted in the manuscript.
